

# A simple model of a growing tumour

David Orrell[1,*] and Hitesh B. Mistry[2,*]

[1] Systems Forecasting, Toronto, Ontario, Canada
[2] Division of Pharmacy, University of Manchester, Manchester, United Kingdom
[*] These authors contributed equally to this work.

## ABSTRACT

This paper presents the CellCycler, a model of a growing tumour which aims to simulate and predict the effect of treatment on xenograft studies or in the clinic. The model, which is freely available as a web application, uses ordinary differential equations (ODEs) to simulate cells as they pass through the phases of the cell cycle. However the guiding philosophy of the model is that it should only use parameters that can be observed or reasonably well approximated. There is no representation of the complex internal dynamics of each cell; instead the level of analysis is limited to cell state observables such as cell phase, apoptosis, and damage. We show that this approach, while limited in many respects, still naturally accounts for a heterogenous cell population with varying doubling time, and closely captures the dynamics of a growing tumour as it is exposed to treatment. The program is demonstrated using three case studies.

## INTRODUCTION

The CellCycler is an ODE-based model of a growing tumour, that is available as a web application. The model uses a novel approach to capture the cell population dynamics as individual cells grow and progress through the cell cycle, and can be used to plot overall tumour growth in response to the effects of anti-cancer drugs. The model is intended as a tool to help researchers simulate existing data, understand the effects of treatment on tumour dynamics, and make basic predictions.

While as discussed in the next section more sophisticated modelling tools exist, the CellCycler was designed to be as simple as possible in terms of the number of parameters (as opposed to the number of equations), while capturing the main necessary features. For example, there is no attempt made to model the complex dynamics of the cell interior. One reason is that these dynamics are difficult to measure, but more important is the fact that we are only interested in how they are expressed through cell growth, arrest, or death, and these effects can be adequately parameterised at the level of the cell. (In other words, the level of analysis here is the cell, or actually groups of cells, rather than the interior of the cell.) If, for example, a drug creates cell damage during a particular phase, then it is usually isn't necessary to simulate exactly how the damage occurs, because what counts here is the amount of cell damage, which can be parameterised in a straightforward manner.

Corresponding author
David Orrell,
dorrell@systemsforecasting.com

It can also be argued that introducing too many parameters actually makes a model less useful, both because of the problem of over-fitting, and because models tend to become unstable as they become increasingly intricate. Indeed the empirical evidence shows that simple models tend to do better in predictive tests (*Makridakis & Hibon, 2000*). Finally, keeping the model simple also makes it much easier and faster to use, and can help make the results easier to interpret. For example, the program could be used as a pedagogical tool to explore the dynamic effects of tumour treatment. The CellCycler therefore is limited to basic parameters that can be observed or approximated reasonably accurately.

Of course, given the complexity of biological systems, simple models like the CellCycler can only ever provide a partial approximation to the underlying dynamics, and can be misleading if used in situations where their assumptions break down. If further detail is desired, and adequate data exists, then it is possible to extend the model, but this should only be done if it is consistent with the model assumptions and adds some predictive value. Otherwise it may make more sense to adopt a more flexible agent-based approach as discussed below.

The paper begins by deriving the basic equations for the different components of the model, including the mathematical treatment of cell population behaviour and tumour growth. An interesting feature of the model is the way that it naturally accounts for a spread in tumour cell doubling times. We then explain how the CellCycler web application is run. Finally, we demonstrate how the CellCycler can be used for exploring combination schedules through three examples. The first example explores a hypothesis around combining two drugs, where one affects cells in G1 phase and the other in M phase. The other two case studies look in a more quantitative manner at the predictivity of the CellCycler for preclinical combination experiments.

## MATERIALS AND METHODS

### Cell population model

The CellCycler model consists of three separate components: a cell population model, a PK model, and a tumour growth model. The cell population model, which is the most complex part of the CellCycler, needs to reflect the growth and phases of the cells as they progress through the cell cycle, and effects such as cell damage and death. It also needs to allow for a spread of cell cycle times within the population.

A number of approaches to modelling a cell population can be found in the literature. One is to model cells individually, using an agent-based model (*Bayrak et al., 2016*). This technique is the most general and powerful since it offers the ability to track and tailor the dynamics of individual cells, but can be complicated and is computationally rather slow. Use and interpretation of such models also often requires significant training and experience.

Another method is to use ordinary differential equations, where each equation models cells in a particular state (*Checkley et al., 2015*). For example there may be separate equations for cells in G1 phase, or S phase, or damaged cells. However such models may not accurately capture the time-dependent effect of certain drugs, which can be important when exploring

phenomena such as synchronisation. For example, if a drug damages cells in S phase, then to determine the effect of the drug on cells, we need to know exactly where cells are in the cycle: if they are just starting S phase when exposed to drug, then they will get more exposure than cells which are about to exit.

The CellCycler model addresses this problem by dividing the cell cycle up into a large number of separate compartments of equal length. As discussed further below, typically $N = 50$ compartments have been found to give adequate resolution. (Note this number does not affect the number of parameters, which are the same for each compartment.) In the absence of drug, the equation for the volume $V_n$ of each compartment is very simple:

$$\frac{dV_n}{dt} = k_1 V_{n-1} - k_2 V_n.$$

Here $k_1$ represents the change in volume due to cells transiting from compartment $n-1$ to compartment $n$, and $k_2$ represents the change in volume due to cells transiting from compartment $n$ to compartment $n+1$ (note the indices are cyclic, so $V_n = V_{N+n}$).

As shown in the Appendix, the rate constants are closely approximated by the formulas

$$k_1 \cong \frac{(N + \log(2))}{t_d}, k_2 \cong \frac{N}{t_d}.$$

The rate constants therefore simply reflect the fact that the cells pass through $N$ compartments in time $t_d$, with an additional growth term appearing in $k_1$ which accounts for cell growth.

The initial condition assumes that each compartment has an identical volume of cells, which is consistent with the assumption that cells grow at a constant rate. For example, if mitosis occurs in the last compartment, then there will be twice as many cells in the first compartment, but the same volume.

## Discretisation effects

Because the CellCycler divides the cell cycle into a fixed number of compartments, one consequence is that the effective doubling time has a degree of uncertainty. This is illustrated by Fig. 1, which shows how a perturbation at time zero in one compartment tends to blur out over time, for models with $N = 25$, 50, and 100 compartments, and a doubling time of $t_d = 24$ hours. In each case a perturbation of size $\frac{N}{t_d}$ is made to the volume of compartment 1 at the beginning of the cell cycle (note that each compartment has a volume which varies inversely with $N$, so this scaling ensures that the perturbation represents the same change in volume for any choice of $N$). For the case with 50 compartments, the curve is closely approximated by a normal distribution with standard deviation of 3.5 h or about 15 percent (dashed green line).

The initial perturbation to a single compartment therefore spreads out with time, in a manner which depends on the number of compartments $N$. This blurring effect due to discretisation is a desirable feature (if it didn't exist, we would have to add it) because it is equivalent to saying that the cell population has a variable doubling time, as is the case in growing tumours. One implication is that synchronisation effects caused by drugs reduce over time. While we don't usually have exact data on the spread of doubling times

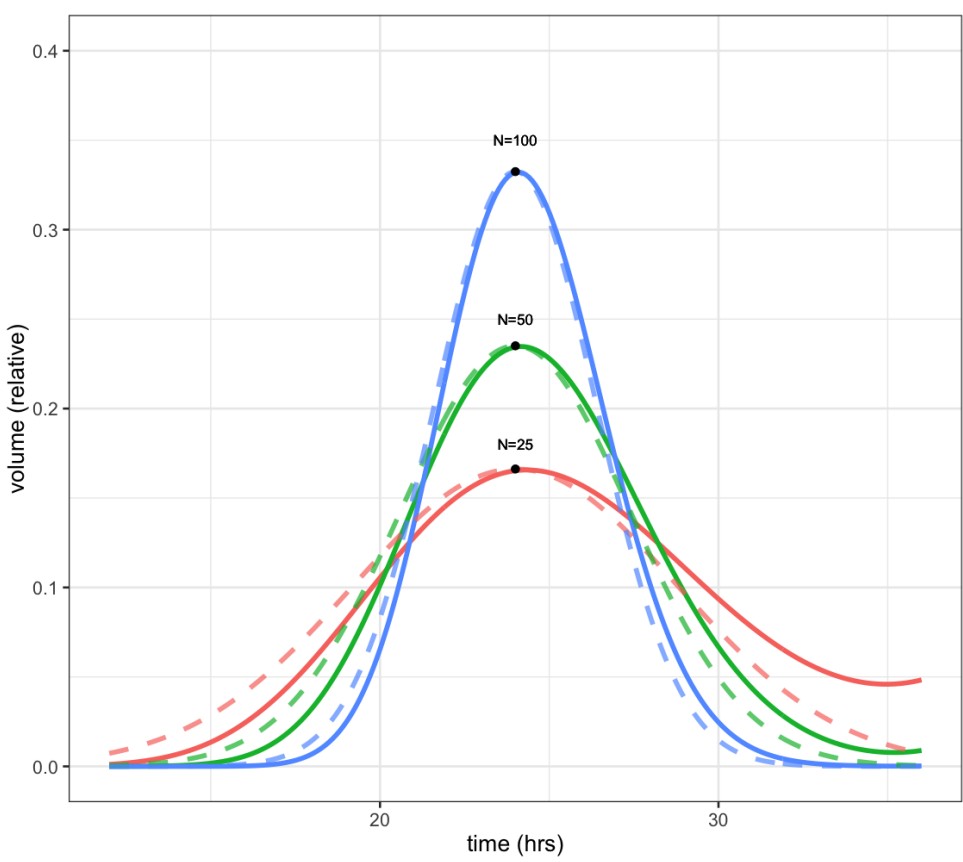

**Figure 1** **Plot showing volume in compartment 1 (of $N$) following a perturbation to that compartment alone, after one cell doubling period of 24 h.** The cases shown are with $N = 25, 50,$ and 100 compartments. The black dots show the estimated peak volumes for the three cases, using the formula developed below. Dashed lines are the corresponding normal distributions. The distribution becomes more concentrated as the number of compartments increases.

in the growing layer, a choice in the region of 50 compartments gives what appears to be a reasonable degree of spread.

It is possible to obtain a very simple analytic expression for the curves shown in the figure. As shown in the Appendix, the peak of the distribution, shown by the black dots in Fig. 1, is given by

$$V_1(t_d) \cong \frac{1}{t_d}\sqrt{\frac{2N}{\pi}}.$$

Note this will increase only slowly with the square root of $N$. It is therefore impractical to obtain a highly peaked distribution; however as mentioned above this would be unrealistic because it would imply a near-perfect degree of cell synchronisation which does not occur in growing tumours.

The distributions in the figure can be approximated by a normal distribution (shown by dashed lines), with peak given by the above formula, and a corresponding standard

deviation, normalised to doubling time, of

$$\sigma = \frac{1}{\sqrt{N}}.$$

So for example using 25 compartments gives a normalised standard deviation of about 0.02 or 20 percent, while using 100 compartments decreases this to 0.01 or 10 percent. The default value is 50 compartments, which gives a standard deviation in doubling times of about 14 percent.

### Effect of drug

When drug is present, some cells will be damaged and possibly repaired, or killed outright. To model these effects, we include in the model an additional $N$ compartments for damaged cells, and one additional variable which represents cells lost to apoptosis. The volume of the damaged cells $D_n$ is then given by

$$\frac{dD_n}{dt} = k_d V_n - k_r$$

where $k_d$ is the drug-dependent rate of damage, and $k_r$ is the repair rate. The rate for the volume $A$ of cells lost to apoptosis is given by

$$\frac{dA}{dt} = k_a V_n.$$

The equation for proliferating cells is correspondingly modified to give

$$\frac{dV_n}{dt} = k_1 V_{n-1} - k_2 V_n - k_d V_n + k_r - k_a V_n.$$

Since $k_1$ and $k_2$ are determined from the cell doubling time, the only additional parameters required by the model are the drug-dependent properties $k_d$, $k_r$ and $k_a$, as well as the allocation of compartments between the different phases.

### Tumour growth

Tumour growth is caused by the proliferation of dividing cells. For example, if cells have a cell cycle length $t_d$, then the total number of cells will double every $t_d$ hours, so the volume will be given by

$$V = 2^{\frac{t}{t_d}} = e^{at}$$

where

$$a = \frac{\log(2)}{t_d}.$$

However, in general only the cells in the periphery of the tumour will be growing, since cells in the inner core do not have access to the necessary nutrients.

Following a number of other models, we therefore assume for simplicity that tumour growth is driven by an outer layer of proliferating cells, surrounding a quiescent or necrotic core. The computational details were presented in a previous paper, but we recap the

argument here (*Mistry, Orrell & Eftimie, 2018*). If the proliferating layer has thickness $d < r$, and is growing at a rate $a$, then the volume of the layer is

$$V_p = 4\pi r^2 d$$

and it is growing at a rate

$$\frac{dV_p}{dt} = aV_p = a4\pi r^2 d.$$

The total volume of the tumour is

$$V = V_p + V_c$$

where $V_c$ is the volume of the necrotic core. The growth equation for the radius of the whole tumour is given by

$$\frac{dr}{dt} = \frac{dr}{dV}\frac{dV}{dt} = \frac{dr}{dV}\left(\frac{dV_p}{dt} + \frac{dV_c}{dt}\right)$$

but since

$$\frac{dV_c}{dt} = 0$$

the growth equation of the radius becomes

$$\frac{dr}{dt} = \frac{dr}{dV}\frac{dV}{dt} = \frac{dr}{dV}\frac{dV_p}{dt} = \left(\frac{1}{4\pi r^2}\right)a4\pi r^2 d = ad$$

which is solved to give the linear equation

$$r = R_0 + adt.$$

To translate from cell population growth (with growth rate $a$) to tumour growth, we therefore need just two additional parameters, which are the thickness of the growing layer $d$, and the initial volume $R_0$. Note that the equation assumes $d < r$ so holds only when the tumour is sufficiently large that it has developed a non-growing core.

This growth equation, which is not new but has been known since at least the 1930s, is consistent with the empirical observation that in the absence of treatment tumour diameter tends to increase in a roughly linear fashion (*Mayneord, 1932*). The model will of course not be a perfect fit for the growth of all tumours, but has the advantage that it can be easily parameterised and fit to noisy data. It can also be extended to more complex cases, for example where drug resistance leads to a modified growth rate after treatment.

## Using the CellCycler

The CellCycler model has been incorporated into a freely accessible Shiny web application (*Orrell & MIstry, 2019*). The starting point for the program is the Cells page, which is used to model the dynamics of a growing cell population. The key parameters are the average cell doubling time, and the fraction spent in each phase (G2 is set automatically since the proportions must add to 1). The doubling time is assumed to be variable, with a range that depends on the number of model compartments. This can be adjusted in the Advanced tab:

25 compartments gives a standard deviation for cell doubling times of about 20 percent, 50 compartments gives 14 percent, and 100 compartments gives 10 percent. Note that the number of compartments affects both the simulation time (more compartments is slower), and the discretisation of the cell cycle. For example with 50 compartments the proportional phase times will be rounded off to the nearest 1/50=0.02.

In addition the user selects the simulation time, and plotting choices such as growing or damaged cells. The plot will then show the volume of cells in each phase, as well as the total volume, normalised to an initial volume of 1. Model settings can be saved to or loaded from a csv file.

The next pages, PK1 and PK2, are used to parameterise the PK models and drug effects. The program has a choice of three PK model types. The first is a simple decay model (K-PD), where the drug is introduced at a certain concentration (as in intravenous bolus injection) and then decays. The second is a step model, where the drug is assumed to be held at a fixed level over specified time intervals, as in infusion. The third option is a one-compartment model which includes absorption and decay rates (a schematic is given in the online documentation—a project for future work is to add other options such as multi-compartment models). In addition the phase of action (choices are G1, S, G2, M, or all), and rates for death, damage, and repair can be adjusted. Units are in terms of free concentration. Finally, the Tumor page uses the model simulation to generate a plot of tumor radius, given an initial radius and growing layer. A table is shown giving total radius gain; the maximum gain that would be obtained in the absence of drug; the radius loss due to drug; and the proportions of this loss that are due to death or cell damage.

The results can be compared with experimental results in the Tumor page by using the "Read data for overlay from file" option, and checking the "overlay" box. A sample file, consisting of data from an adenoid cystic carcinoma study, is currently loaded by default for demonstration purposes (*Moskaluk et al., 2011*). If the user selects "show linear fit" then the plot will include a linear interpolation, with estimates for initial volume and growing layer thickness given in a table. If the data corresponds to control curves with no drug, then these values can be used to parameterise the growth rate of the model.

## RESULTS

In the following sections, we present three case studies which show how the CellCycler can be applied in drug combination studies. The first is a fairly qualitative analysis of a drug combination, while the next two show the CellCycler's use as a predictive tool.

### Case Study 1: EGFR/Chemotherapy combinations

EGFR inhibitors are now widely used within the clinic as a monotherapy agent for treating patients with non-small cell lung cancer who harbour a specific EGFR mutation. During their clinical development, however, EGFR inhibitors were also studied in combination with chemotherapy within the lung cancer setting. The results of the combination were not as efficacious as hoped given the strong preclinical results. One explanation has centred on the antagonism between the two drugs at the cell-cycle level (*Davies et al., 2006*). EGFR inhibitors are known to exert their effect during G1 phase of the cell cycle, whereas the

**Table 1  EGFRi/Chemotherapy combination study.** Parameter values used for the EGFR inhibitor in combination with Chemotherapy example.

| Parameter | Value |
| --- | --- |
| Cell-Cycle (Default Settings) | |
| Doubling Time (hours) | 24 |
| Cell-Cycle Phases | |
| G1 | 0.2 |
| S | 0.3 |
| G2 | 0.4 |
| M | 0.1 |
| Tumour Size (Default Settings) | |
| Initial Diameter (mm) | 6.8 |
| Growing Layer (mm) | 0.24 |
| Drug PK (Default Settings) | |
| EGFR Inhibitor | |
| Dose/V | 1 |
| Elimination Rate | 0.1 |
| Taxol | |
| Dose/V | 1 |
| Elimination Rate | 0.1 |
| Drug Cell-Cycle (Default Settings) | |
| EGFR Inhibitor | |
| G1 Phase Damage | 1 |
| G1 Phase Repair | 0.1 |
| G1 Phase Apoptosis | 0 |
| Taxol | |
| M Phase Damage | 1 |
| M Phase Repair | 0.1 |
| M Phase Apoptosis | 1 |

Taxol based chemotherapy treatments affect cells in M phase. Furthermore, the effect of EGFR inhibitors on G1 phase of the cell-cycle involves arresting the cells in that phase, thus delaying progression through the cell-cycle. If both agents are given at the same time, there could therefore be a degree of antagonism which may reduce the effect of the combination.

An obvious solution to this problem is to sequence the treatments in such a way as to prevent this antagonism. We therefore used the CellCycler to explore how much of a difference sequencing makes, versus giving both agents at the same time, in a hypothetical xenograft experiment.

The parameter values used for this case study can be seen in Table 1. They represent the default settings of the CellCycler. For the two drugs of interest we have used the default settings for the K-PD model. The EGFR inhibitor was modelled as causing damage to cells in G1-phase. For a Taxol based inhibitor we modelled the drug as affecting cells in M-phase, and assumed both cell damage and cell death can occur.
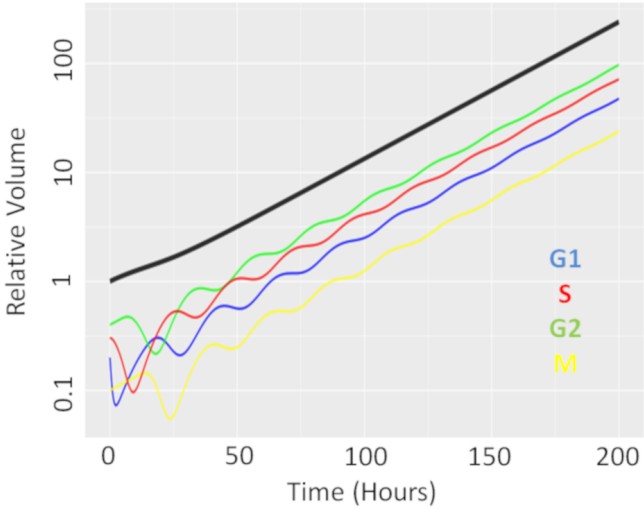

**Figure 2** **Temporal evolution of cell-cycle voumes.** Plot showing how a single dose of an agent effecting cells in G1-phase of the cell-cycle effects the cell-cycle distribution over time.

We first performed a single dose simulation to highlight the gain and loss of cell-cycle synchronisation effects with the CellCycler, see Fig. 2. The plot shows the temporal evolution of the relative volumes in different phases of the cell-cycle after a single administration of the EGFRi. We can see that the proportion of cells in G1 phase of the cell-cycle (blue line) first drop before re-bounding as the drug clears and cells recover from the treatment effect. The simulation highlights that giving a drug that works within M-phase of the cell-cycle (yellow line) at 24 h is not optimal as that is when the proportion of cells in that phase of the cell-cycle is at its lowest.

We then moved onto the repeat dosing studies. The overall simulation time used was three weeks, which is considered a reasonable time for a xenograft experiment. The schedules explored, and the results of the simulations, can be seen in Table 2. We see that the combination effect is indeed sequence dependent. Giving Taxol first, followed by the EGFRi, gives modest improvement over the combination given at the same time. However, reversing the order—with the EGFRi first, followed by Taxol—is less efficacious then the two drugs given together.

These results show that the CellCycler can be used to explore sequencing effects for combination therapies. The next two examples we shall consider are more quantitative ones.

## Case study 2: CDK4/6 inhibitor combination with Gemcitabine

In this example we apply the CellCycler to a combination involving a CDK4/6 inhibitor (LY2835219) and Gemcitabine. The experiment of interest is taken from *Gelbert et al. (2014)* which tested the sequencing effect of the two drugs in a Calu-6 xenograft model (see Fig. 5 in that paper). We will consider three of the experiments as a training set for model parameterisation: control (without drug), LY2835219 as monotherapy (50 mg/Kg once daily for 21 days), and Gemcitabine (60 mg/Kg once every 3rd of a 7 day cycle)

**Table 2 CellCycler simulation results.** Simulation results in terms of diameter loss for the drugs as monotherapy and in combination.

| Drug schedule | Diameter loss |
| --- | --- |
| Taxol | |
| Day 1 weekly | 0.57 mm |
| EGFRi | |
| Days 1–5 every week | 0.90 mm |
| Taxol + EGFRi | |
| Taxol (Day 1) + EGFRi (Days 1–5) weekly | 1.38 mm |
| Taxol ->EGFRi: | |
| Taxol (Day 1) + EGFRi (Days 2–6) weekly | 1.44 mm |
| EGFRi ->Taxol: | |
| EGFRi (Days 1–5) + Taxol (Day 6) weekly | 1.31 mm |

as monotherapy. These studies together with literature information on Calu-6 doubling time (*Lemaire et al., 2008*) and cell-cycle phase durations (*Han et al., 2008*) were used to calibrate all the parameters in the model; see Table 3. We assumed that LY2835219 exerts its effect on cells when in G1 phase of the cell-cycle, this is consistent with what is known about the compound from Gelbert et al. Gemcitabine is well known to cause damage to cells in S phase of the cell-cycle which can lead to apoptosis, and so we model the drug in this way. The model fits to the training data, obtained by adjusting parameters until the simulated growth curve adequately agreed with the data, can be seen in Fig. 3.

It is noticeable in Fig. 3 that the CellCycler is underestimating the diameter curves after the treatment has stopped, post day 21. However, we are using the mean value at each experimental time-point and so this must be taken into account when judging the model fits. Individual mice data were not available in the literature. The CellCycler can be considered to be well-calibrated to the experimental data up to the last dosing day, day 21, for the treated xenografts which was our aim.

We then simulated the dosing schedules for the combination schedules, i.e., sequenced (LY first: 50 mg/Kg once daily for 12 days; then Gemcitabine: 60 mg/Kg every 3rd day of a 4 day cycle) versus both compounds taken together (LY, 50 mg/Kg once daily for 21 days, and Gemcitabine, 60 mg/Kg every 3rd day of a 7 day cycle), which formed our testing set. The results can be seen in Fig. 4. The CellCycler predictions for the testing sets are in good agreement with the experimental data. The results show that there is little difference in terms of efficacy between the two dosing schedules, even though the schedules are quite different.

## Case study 3: MEK inhibitor combination with Docetaxel

In this example we apply the CellCycler to a combination involving a MEK inhibitor (Selumetinib) and Docetaxel. The experiment of interest is taken from Holt et al. which tested the sequencing effect of the two drugs in a HCT-116 xenograft model (see Fig. 3C in that paper) (*Holt et al., 2012*). Similar to case study 2 we will consider three of the experiments as a training set for model parameterisation: control (without drug), Selumetinib as monotherapy (twice daily for 7 days), and Docetaxel as monotherapy

**Table 3  CDKi/Gemcitabine combination study.** Parameter values for the CDK inhibitor in combination with Gemcitabine example.

| Parameter | Value |
|---|---|
| Cell-Cycle (Default Settings) | |
| Doubling Time (hours) | 30 |
| Cell-Cycle Phases | |
| G1 | 0.5 |
| S | 0.25 |
| G2 | 0.15 |
| M | 0.1 |
| Tumour Size (Default Settings) | |
| Initial Diameter (mm) | 6.32 |
| Growing Layer (mm) | 0.46 |
| Drug PK (Default Settings) | |
| LY2835219 | |
| Dose/V | 1 |
| Elimination Rate | 0.1 |
| Gemcitabine | |
| Dose/V | 1 |
| Elimination Rate | 0.1 |
| Drug Cell-Cycle | |
| LY2835219 | |
| G1 Phase Damage | 0.4 |
| G1 Phase Repair | 0.1 |
| G1 Phase Apoptosis | 0 |
| Gemcitabine | |
| S Phase Damage | 10 |
| S Phase Repair | 0.1 |
| S Phase Apoptosis | 0.8 |

(single dose). These studies were combined with literature information on HCT-116 doubling time (*Li, Nelsen & Hendrickson, 2002*) and cell-cycle phase durations (*Hemmati et al., 2005*) to calibrate all the parameters in the model; see Table 4. We assumed that Selumetinib exerts its effect on cells when in G1 phase of the cell-cycle, this is consistent with what is known about the compound (*Holt et al., 2012*). Docetaxel is known to cause damage to cells in M phase of the cell-cycle which can lead to both cell damage and apoptosis, and so we model the drug in this way.

The model fits to the training data, obtained by adjusting parameters until the simulated growth curve adequately agreed with the data, can be seen in Fig. 5. The control data curve appears to grow linearly until the last time-point at which the mean value drops. This could well be due to the effect of drop-outs. Individual mice growth curves would be needed to account for this. We therefore chose to fit to the earlier time-points with the final data-point ignored. The model fit to Selumetinib is in good agreement with the mean

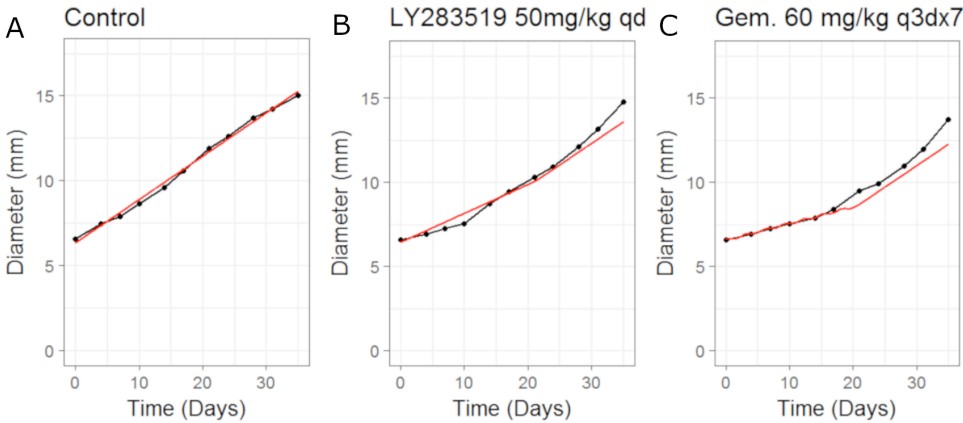

**Figure 3** **Plot showing the calibration of the CellCycler to the control (A) and monotherapy data (B and C).** Simulation (red line) overlaid on top of the experimental data (black line) used for training the CellCycler. The original volume data has been converted to diameter values through a cube-root transformation. The original volume data has been converted to diameter values through a cube-root transformation.

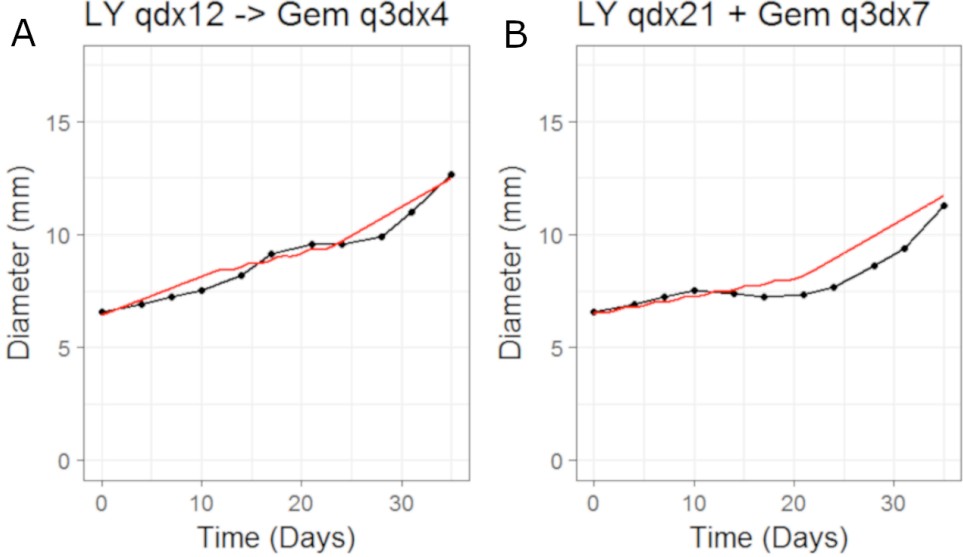

**Figure 4** **Plot showing the prediction of the combination studies using the CellCycler.** Simulation (red line) overlaid on top of the experimental data (black line) used for testing the CellCycler. (A) shows LY283519 (LY) given first followed by Gemcitabine (Gem) whereas (B) shows LY given in combination with Gem up-front. The doses of LY and Gem were the same as that used for monotherapy studies. The original volume data has been converted to diameter values through a cube-root transformation.

data. The fits to Docetaxel data are not as good, however again it must be noted that we are using mean data.

We then simulated two different combination schedules: (i) Selumetinib given first, twice daily for 7 days, then after 24 h of the last dose of Selumetinib a single dose of

**Table 4  MEKi/Docetaxel combination study.** Parameter values used for the MEK inhibitor (MEKi) in combination with Docetaxel example.

| Parameter | Value |
|---|---|
| Cell-Cycle (Default Settings) | |
| Doubling Time | 18 |
| Cell-Cycle Phases | |
| G1 | 0.49 |
| S | 0.33 |
| G2 | 0.09 |
| M | 0.09 |
| Tumour Size (Default Settings) | |
| Initial Diameter (mm) | 8.30 |
| Growing Layer (mm) | 0.50 |
| Drug PK (Default Settings) | |
| MEKi | |
| Dose/V | 1 |
| Elimination Rate | 0.1 |
| Docetaxel | |
| Dose/V | 1 |
| Elimination Rate | 0.0025 |
| Drug Cell-Cycle | |
| MEKi | |
| G1 Phase Damage | 0.1 |
| G1 Phase Repair | 0.001 |
| G1 Phase Apoptosis | 0 |
| Docetaxel | |
| M Phase Damage | 0.5 |
| M Phase Repair | 0.001 |
| M Phase Apoptosis | 0.1 |

Docetaxel was given; ii) single dose of Docetaxel then after 24 h 7 days of twice daily dosing of Selumetinib. The simulations overlayed on top of the experimental data can be seen in Fig. 6. Schedule (i) is in good agreement with the experimental data with a slight under prediction. Schedule (ii) is well predicted until the first data-point after which we over-predict the tumour volume. To better understand the difference between model prediction and mean observation, individual mice growth curves would be necessary. Although the quantitative predictions were not as good as those for case study 2, the qualitative result that there is a modest difference between the two schedules observed in the experiment was also observed in the model predictions.

## DISCUSSION

In summary, the three case studies presented highlight the potential of the CellCycler to be used to explore hypotheses and also as a predictive tool. The model is conceptually simple, and the minimal number of parameters mean that it is both less vulnerable to over-fitting,
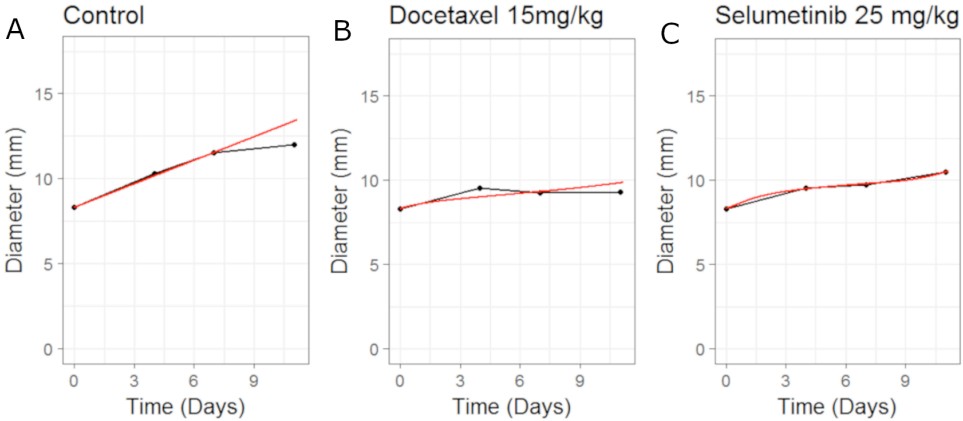

**Figure 5** **Plot showing the calibration of the CellCycler to the control (A) and monotherapy data (B and C).** Simulation (red line) overlaid on top of the experimental data (black line) used for training the CellCycler. The original volume data has been converted to diameter values through a cube-root transformation.

and easier to interpret and compare across cases, than more complicated models. The model may also be of use as a pedagogical tool.

The model uses cells as the units of analysis, and does not attempt to simulate intra-cellular dynamics, but can still capture effects such as cell death or damage in a straightforward way. Three case studies were explored which highlight how the sequence in which drugs are used can affect the tumour growth time-series. The core model can also be extended in a number of ways to include specific checkpoints, or to capture effects such as drug resistance or growth saturation. However as a rule this should only be done if it makes the model simulations more accurate and predictive (rather than simply for completeness) since additional parameters make the model more prone to over-fitting. Readers are invited to try out the program for themselves by accessing it online, either with the sample data set provided or with their own data.

## APPENDIX

### Derivation of rate equations

We assume that cells grow constantly as they progress through the cell cycle. As cells enter from the previous compartment, their volume therefore expands by a factor $2^{\frac{1}{N}}$. So after passing through all $N$ compartments, the cells will have grown by a factor 2 as expected (after which they divide in two). It follows that the volume change rate $k_1$ corresponding to cells entering from the previous compartment must be a factor $2^{\frac{1}{N}}$ greater than the corresponding rate $k_2$ for cells leaving the current compartment, i.e., $k_1 = 2^{\frac{1}{N}} k_2$.

If we now sum the rate equations for the separate compartments, we obtain the growth rate for the total volume $V_T$ which is

$$\frac{dV_T}{dt} = (k_1 - k_2) V_T.$$

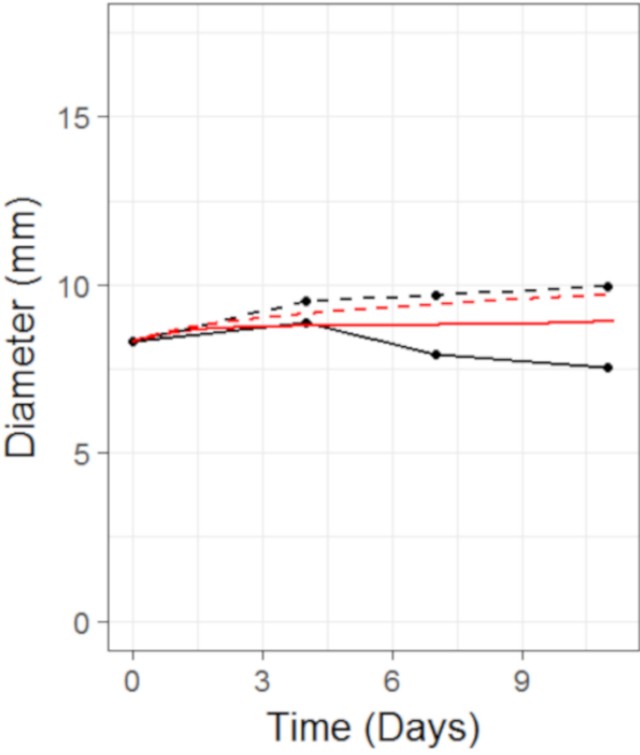

**Figure 6** **Plot showing the prediction of the combination studies using the CellCycler.** Simulation (red line) overlaid on top of the experimental data (black line) used for testing the CellCycler. The dashed lines represent Selumetinib (MEKi) given first followed by Docetaxel (DOC) whereas the solid line represents DOC given first and then the MEKi. The doses of MEKi and DOC were the same as that used for monotherapy studies. The original volume data has been converted to diameter values through a cube-root transformation.

If the average length of the cell cycle is $t_d$, then the total number of cells is doubling in time $t_d$, and we can also express the growth rate as an exponential equation

$$\frac{dV_T}{dt} = e^{at}$$

where

$$a = \frac{\log(2)}{t_d}$$

We therefore have two equations for the two unknowns $k_1$ and $k_2$, which can be solved to give

$$k_1 = \frac{\log(2)}{t_d} \frac{2^{1/n}}{2^{1/n}-1}, k_2 = \frac{\log(2)}{t_d} \frac{1}{2^{1/n}-1}$$

Using the approximation, valid for large $n$, that

$$\frac{2^{1/n}}{2^{1/n}-1} \cong \frac{1}{\log(2)}$$

we note that the rates are roughly

$$k_1 \cong \frac{(N + \log(2))}{t_d}, k_2 \cong \frac{N}{t_d}.$$

## Derivation of approximate formula for doubling time variability

The model equations for growing cells can be written:

$$\frac{d\mathbf{V}}{dt} = (k_1 \mathbf{J} - k_2 \mathbf{I})\mathbf{V}$$

where $\mathbf{I}$ is the identity matrix, and $\mathbf{J}$ is the identity matrix with columns permuted once to the left. For example, for the case with only four compartments, we would have

$$\mathbf{J} = \begin{pmatrix} 0 & 0 & 0 & 1 \\ 1 & 0 & 0 & 0 \\ 0 & 1 & 0 & 0 \\ 0 & 0 & 1 & 0 \end{pmatrix}.$$

This equation has as its solution

$$\mathbf{V} = e^{(k_1 t \mathbf{J} - k_2 t \mathbf{I})}\mathbf{V_0} = e^{-k_2 t}e^{k_1 t \mathbf{J}}\mathbf{V_0}.$$

To determine the effect after one doubling time, we use the approximations for $k_1$ and $k_2$ above, and set $t = t_d$ to obtain

$$k_1 t_d \cong N + \log(2), k_2 \cong \frac{N}{t_d}$$

so

$$\mathbf{V} \cong e^{-N}e^{(N + \log(2))\mathbf{J}}\mathbf{V}(0)$$

Taking a Taylor expression, we have

$$e^{(N + \log(2))\mathbf{J}} = \mathbf{I} + (N + \log(2))\mathbf{J} + \frac{1}{2!}(N + \log(2))^2 \mathbf{J}^2 + \frac{1}{3!}(N + \log(2))^3 \mathbf{J}^3 + \dots$$

Consider a step perturbation so that the vector $\mathbf{V(0)}$ is 0 in all elements except for the first, normalised for the total volume being perturbed so that

$$V_1(0) = \frac{N}{t_d}.$$

In this case all powers of the matrix $\mathbf{J}$, when multiplied by $\mathbf{V(0)}$, are zero except for those powers that are divisible by $N$, so the expression for the volume $V_1$ in the first compartment after time $t_d$ reduces to

$$V_1(t_d) \cong e^{-N}\left(1 + \frac{1}{N!}(N + \log(2))^N\right)\frac{N}{t_d}$$

plus higher terms which can be neglected over time scales of a single cycle.

Using Stirling's approximation

$$N! \cong \sqrt{2\pi N}\left(\frac{N}{e}\right)^N$$

then gives

$$V_1(t_d) \cong \frac{1}{\sqrt{2\pi N}}\left(\frac{N + \log(2)}{N}\right)^N \frac{N}{t_d}.$$

The term in brackets converges for large $N$ to 2 (this can be seen by taking the logarithm and using a first-order approximation), so

$$V_1(t_d) \cong \frac{1}{t_d}\sqrt{\frac{2N}{\pi}}.$$

### Funding
The authors received no funding for this work.

### Competing Interests
David Orrell and Hitesh B. Mistry are cofounders of Systems Forecasting.

### Author Contributions
- David Orrell and Hitesh B. Mistry conceived and designed the experiments, performed the experiments, analyzed the data, contributed reagents/materials/analysis tools, prepared figures and/or tables, authored or reviewed drafts of the paper, approved the final draft.

### Data Availability
Data is available at GitHub: https://github.com/Orrell/CellCycler.

### Supplemental Information
Supplemental information for this article can be found online at http://dx.doi.org/10.7717/peerj.6983#supplemental-information.

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
