# Peer review of "A simple model of a growing tumour"

_PeerJ, doi:10.7717/peerj.6983_

## Round 0.1 · original submission · Major Revisions

Many thanks for submitting your manuscript to PeerJ. I have now received comments from 2 reviewers with expertise in the field. They have each raised considerable concerns, which you will see in their attached comments.

I hope you will appreciate that the reviewer comments are intended to be helpful and aim to improve the value of your submission.

If you are able to address each and every issue raised by the reviewers, we would welcome you to submit a revised version of your manuscript. I encourage you to incorporate changes into the revised manuscript, not solely in a rebuttal letter - potential readers of the manuscript will not necessarily be aware of the rebuttal letter and will only see the final manuscript, which therefore needs to be independent of any of your responses to the reviewers and must clearly explain all aspects to the reader, not just the reviewers, editors and yourselves.

·

Basic reporting

The paper is clearly written and presented, with appropriate referencing.

Experimental design

In general the tumour model seems clear. However, I didn't follow the description of the PK models. Eg I'm not entirely clear what the authors mean by 'simple decay (K-PD), a step model, or a one-compartment model'. Some elaboration of the models proposed, the mathematical description and the extraction of the parameters from xenograft experiments for the evaluation would be helpful.

Validity of the findings

The investigation of the combinations and dose ordering effects is an interesting aspect of the manuscript.

Additional comments

Although I think the work is good quality, I'm not clear what it adds scientifically. The model looks like a hybrid Simeoni/cell cycle model with perhaps some simplifications. Both of these are already described in the literature and hence this paper is not a major step forward in tumour modelling, in my opinion. I'm not sure about the value of linking the paper to a simulator is, unless the model can be accessed and edited? Eg if the authors could publish the code in an open source software, then I think that this may be of utility and interest.

Reviewer 2 ·

Basic reporting

Orell and Mistry ms describes a mathematical model of a growing tumour to simulate and predict the effect of treatment on xenograft studies and potentially translate them to the the clinic. Their model uses parameters that can be observed or reasonably well approximated which is a strong argument.

The English is in general good, although in several instances it could do with more formality and/or clarity. (e.g. line 171 “we recap the argument here”, please review)

The introduction and background are weak. There is no real background given in the introduction to understand what other models have been developed in the past, how useful these have been, and what are in the authors opinions the pros and cons of each method. There is some background given in the methods but it’s really little and not placed appropriately.

The literature cited is relevant, but maybe again on the light side.

The figures could be improved (thicker lines, larger labels). Also, it ismy opinion that the ms requires more figures (see general comments below).

The data that has been used comes from published articles. It has not been provided here as .xls or other format.

Experimental design

The primary research question appears to be the possibility of predicting combination xenograft experiments based on control and monotherapy data, using knowledge of cell-cycle dependent drug effects.

It is not clear though how the research fills an identified knowledge gap due to the almost absent literature review and discussions.

The investigation was performed to a good technical standard but it will benefit from more elaboration at all levels.

Methods are described appropriately, although several details are missing (see general comments below).

Validity of the findings

The findings appear valid to this reviewer.

Additional comments

Regarding agent-based modelling approaches, the authors should note that these cannot be easily parametrized using available data.

Line 81. The authors refer to 2 other ODEs approach but highlight that “such models may not accurately capture the time-dependent effect of certain drugs”. It’s not clear why this is stated, as the referenced papers are ODEs system where the cell cycle is modelled, and drug targets are also cell cycle dependent.

Line 87. Similar comments. It is stated “The CellCycler model addresses this problem by dividing the cell cycle up into a number of separate compartments of equal length”. I believe this is what have also been done in at least one of the referenced papers.

I assume “figure below” (l.110) is figure 1, in which the volume in compartment 1 is shown following a perturbation to that compartment alone after one cell doubling period of 24 hours. It is not clear to me what type of perturbation has been introduced. It is said “the magnitude is scaled to the number of compartments, so the total size of the perturbation is the same in terms of total volume”. Please be more mathematically explicit.

What do the authors mean by “blurring effect”? Why is this “equivalent to saying that the cell population has a variable doubling time”? It is difficult to comment on what follow without these details.

Regarding drug effect, “repairable” and “apoptotic” states are introduced. A diagram of the proposed model would help. Have the author analysed the effect of modelling these as parallel (r-Main-a), rather than concatenated (Main-r-a)? Why this choice?

Please review or elaborate a bit more, as it results to me that if V=4 pi r2 d, then dr/dt=1/(2r) not 1/(4 pi r2).

Please remind that the resulting growth equation is only valid once the tumor has grown enough and a necrotic core has developed. What would be appropriate R0 size for that to hold? Please elaborate on how a scientist would assess that for his/her xenograft model.

Similarly, please explain/discuss how the user would chose typical fractions for G1, S and M phases.

Important information (e.g. the schematic of the compartment PK models) are given in the online documentation. These should be incorporated in the paper supplementary material.

When referring to Dose/Volume (l.211), please precise this is the drug volume of distribution, to avoid confusion with tumur outer layers volume and/or cell populations volumes.

Please note that the drug concentration in the tumour is in many cases different from plasma pk – the PK models. The simple decay or 1-compartment PK models implemented here are not appropriate in these cases. It would be good to consider adding at least a 2 (and maybe even 3) compartment models.

In practice, how would a user decide on apoptosis, damage and repair rates? I would imagine that caspases, gamaH2AX and repairs biomarkers should be used. However, in practice, the information available is only based on proliferation assays which only display the aggregate effects of these rates. Please comment and elaborate.

How are the “schedule” and “week repeated” entries in the shiny app defined?

l.219 “The results can be compared with experimental results by using the “Read data from file” option.” I could not find this option/button in the app??

l.220 A sample file, consisting of data from an adenoid cystic carcinoma study, is currently loaded by default for demonstration purposes.” As mentioned, could not find it. Please also provide some background information about the data.

l.222 “If the user selects “show linear fit” then the plot will include a linear interpolation, with estimates for initial volume and growing layer thickness given in a table.” What type of data is being referred to? Tumor growth data? What is being estimated exactly, the initial tumor volume? How is the growing layer thickness defined then? Please give details of interpolations and fitting procedures.

l. 255 “The overall simulation time was three weeks, which is the typical length of a xenograft
experiment.” 3 weeks is a good average duration but not typical. There are many instances of longer (months) and even shorter experiments. Maybe change to “often used” or else instead.

The repair rate is fixed for both drugs – I appreciate the choice in the absence of information at this level, however, is there really evidence that repair rates in different phases are the same?

Case study 1: I assume the apoptosis rate for the EGFRi is set to 0.0 (not present in table 1)?

Plots of the results with all the details about the simulation parameters should be included to enable users to understand and replicate them. At the moment table 1 does not contain all the entries requires in the app and table 2 gives some summary results but there are no plots of the simulations.

The case is not discussed enough. There are only 2 sentences starting on l.257 and more mechanistic insights, together with plots of population dynamics, should be given to understand the different regimens results.

Case study 2:

Again, shall the reader assume that the apoptosis rate for the CDK4/6i is set to 0.0 (not present in table 3)?

Figure 2 shows fitting of the tumour radius to the data but there are no details of what has been parametrized and how the “calibration” has been performed.

I don’t think that anything is offered in the app for fitting/calibration purposes. Is the expectation that the user would have performed his own model fittings with alternative software prior to simulating the model with CellCycler?

The authors note (l.280) that “It is noticeable in Figure 2 that the CellCycler is underestimating the diameter curves after the treatment has stopped, post day 21. However, we are using the mean value at each experimental time-point and so this must be taken into account when judging the model fits.” Why using mean data would imply difficulties in capturing post-treatment kinetics. Also, standard deviations from the experimental data should be added in Figure 2.

Figure 3 also needs experimental standard deviations.

Case study 3
Most of previous comments apply here too.

Overall, it would have been better to have examples where combinations predictions work and examples where they also don’t work. Mechanistic insights would benefit the manuscript.

Appendix

l.363-365 The phrasing is a bit strange. The authors should simply start from VT=VT0*exp(k1-k2)t being solution to the ode, hence the relationship with the doubling time.

---

## Round 0.2 · Minor Revisions

Many thanks for submitting your revised manuscript, which has now been re-reviewed by the two original reviewers. You will see that one reviewer has commented on specific aspects of the manuscript and its potential limitations, and has made some specific suggestions. The other reviewer (R2) feels that your revisions have not dealt with their original comments sufficiently, and recommends further revision to take comments relating to rigorous scientific reporting into account more thoroughly.

As before, I hope you will see these as positive criticisms aimed to improve the value of your manuscript and I hope you will be able and willing to revise the manuscript accordingly.

·

Basic reporting

No comment

Experimental design

No comment

Validity of the findings

No comment

Additional comments

The authors comment in the introduction that 'keeping the model simple makes the results easier to interpret'. I think this is not necessarily true (eg where the simple model cannot simulate a novel observed dataset and/or give a clear link that gives mechanistic insight) and they should consider acknowledging the weaknesses of overly simple models. They note themselves that '. There are no shortcuts, which is why the CellCycler uses a large number of equations', and it looks to me like the GitHub code is some 700 or so lines of code.The authors note themselves that simple models do not give sufficient insight, elaboration is needed to describe the data they present and in all likelihood further mechanistic detail will be required in the future. So, perhaps the title ought to be 'A more mechanistic model of tumour growth'? Nevertheless, I agree that this tool could be helpful in the education of non-specialists and perhaps preclinical experimental design. Minor suggestions; I think that the K-PD might mislead some of the potential user group. iv bolus would be a more common description, in my opinion. 2 compartment PK is probably the most commonly observed PK profile, so perhaps adding this model would be helpful? Also, the app isn't clear about units; most drug discovery scientists use CL rather that rate constants and so this might cause some confusion. It is not clear to me whether the PK relates to free or total drug; if the latter then free fraction needs to be accounted for. Overall, I think this is good work and it is very positive that the authors intend to publish the code.

Reviewer 2 ·

Basic reporting

No further comments

Experimental design

No further comments

Validity of the findings

No further comments

Additional comments

Previous comments should have been addressed more thoroughly. No further comments

---

## Round 0.3 · accepted · Accept

Thanks for the revisions, which have improved the manuscript

#